# Synthesis and Characterization of High-Purity Mesoporous Alumina with Excellent Adsorption Capacity for Congo Red

**DOI:** 10.3390/ma15030970

**Published:** 2022-01-27

**Authors:** Zhonglin Li, Ding Wang, Fengcheng Lv, Junxue Chen, Chengzhi Wu, Yuping Li, Jialong Shen, Yibing Li

**Affiliations:** 1Department of Materials Science and Engineering, Guilin University of Technology, Guilin 541000, China; dahe121133@gmail.com (Z.L.); dingnvhuang@gmail.com (D.W.); lfc13217836260@gmail.com (F.L.); cchenjunxue@gmail.com (J.C.); wuchengzhi73@gmail.com (C.W.); 6613003@glut.edu.cn (Y.L.); Jialong.Shen@glut.edu.cn (J.S.); 2Collaborative Innovation Center for Exploration of Nonferrous Metal Deposits and Efficient Utilization of Resources, Guilin University of Technology, Guilin 541000, China; 3Key Laboratory of New Processing Technology for Nonferrous Metals and Materials, Ministry of Education, Guilin University of Technology, Guilin 541000, China

**Keywords:** HPMA, direct aging method, ammonium salt substitution method, Congo Red adsorption

## Abstract

We explore a more concise process for the preparation of high-purity alumina and to address the problem of some conventional micro- and nano-adsorbents having difficulty in exposing their adsorption sites to target pollutants in solution due to the heavy aggregation of the adsorbent, which confers poor adsorption properties. The methods of using gamma-phase high-purity mesoporous alumina (HPMA), with its excellent adsorption properties and high adsorption rates of Congo Red, and of using lower-cost industrial aluminum hydroxide by direct aging and ammonium salt substitution were successfully employed. The results showed that the purity of HPMA was as high as 99.9661% and the total removal rate of impurities was 98.87%, a consequence of achieving a large specific surface area of 312.43 m^2^ g^−1^, a pore volume of 0.55 cm^3^ g^−1^, and an average pore diameter of 3.8 nm. The adsorption process was carried out at 25 °C, the concentration of Congo Red (CR) dye was fixed at 250 mg L^−1^ and the amount of adsorbent used was 100 mg. The HPMA sample exhibited an extremely fast adsorption rate in the first 10 min, with adsorption amounts up to 476.34 mg g^−1^ and adsorption efficiencies of 96.27%. The adsorption equilibrium was reached in about 60 min, at which time the adsorbed amount was 492.19 mg g^−1^ and the dye removal rate was as high as 98.44%. One-hundred milligrams of adsorbent were weighed and dispersed in 200-mL CR solutions with mass concentrations ranging from 50–1750 mg L^−1^ to study the adsorption isotherms. This revealed that the saturation adsorption capacity of the produced HPMA was 1984.64 mg g^−1^. Furthermore, the process of adsorbing Congo Red in the synthesized product was consistent with a pseudo-second order model and the Langmiur model. It is expected that this method of producing HPMA will provide a productive, easy and efficient means of treating toxic dyes in industrial wastewater.

## 1. Introduction

High-purity alumina (HPA), one of the cutting-edge materials of the 21st century, is widely used in modern industrial products for its excellent physical and chemical properties, for instance, high strength wear and corrosion-resistant candle materials, special gas additives, advanced ceramic materials, catalysts and their carriers, biomedical, protective materials, single crystal materials, laser crystals and many other fields [1,2,3,4,5,6,7,8,9,10,11]. Impurity elements are mostly present in alumina in the form of oxides, such as sodium oxide, iron oxide, silicon dioxide, etc., and diverse miscellaneous impurities have different effects on its performance, thus limiting the widespread use of alumina. HPA is classified into the 3N (99.9%), 4N (99.99%) and 5N (99.999%) grades, according to the mass purity of the alumina. HPA price and volume demand both depend on its purity [12], as shown in Table 1. Therefore, the study of the preparation process of high-purity alumina has very broad prospects, far-reaching significance and economic impact.

With the progress in and development of science and technology, there have been a series of preparation methods for HPA materials, including organic aluminum alcohol hydrolysis, ammonium aluminum sulfate pyrolysis, ammonium aluminum carbonate pyrolysis, the modified Bayer method and the roasting method, etc. [13,14,15,16,17,18,19,20].

The method of alcohol–aluminum hydrolysis has often been employed in China. In detail, in the presence of a catalyst, metallic aluminum and organic alcohol are mixed and reacted to obtain an alcohol–aluminum solution, which is followed by hydrolysis and high-temperature roasting of the intermediates to synthesize HPA products. The advantage of this method is the high purity and small particle size of the prepared alumina products, whereas, its high production cost and its being a complicated and uncontrollable process limit its economic efficiency.

The pyrolysis of ammonium aluminum sulfate is a traditional preparation method that has been studied by researchers both in China and globally that focuses on controlling the synthesis conditions to obtain pure ammonium aluminum sulfate, or on crystallizing the resulting ammonium aluminum sulfate several times for purification purposes. The process is cheap and is easily obtains the required raw materials, while the parent alcohol can be recycled, reducing the burden of waste liquid disposal. However, the product may not be calcined sufficiently, resulting in significant sulfate content and unsatisfactory purity, and the ammonia and sulfur trioxide produced pollute the environment, which is not conducive to the realization of environmental friendliness.

The ammonium aluminum carbonate pyrolysis method improves the above technology of the pyrolysis of ammonium aluminum sulfate and overcomes the disadvantages of the production process of ammonium aluminum sulfate, which pollute the environment; however, this method increases the burden of (NH_4_)SO_4_ waste liquid disposal, which is also pollutes the environment.

Congo Red (CR), a water-soluble acidic anionic azo pigment, is commonly used in the textile industry because of its strong adsorption to cellulose; thus, the discharge of wastewater in the textile industry contains Congo Red components. It can affect aquatic organisms and the food chain, even if the CR concentration in water is very low [21], such as by increasing chemical oxygen demand (COD) at the water surface and interfering with sunlight penetration through the water surface, reducing photosynthetic activity [22]. In addition, it can seriously affect kidney function, the reproductive system, liver function, the brain and central nervous system and eventually cause allergic reactions [23] to water quality or even cancer. Given China’s emphasis on environmental protection and the gradual increase in national awareness of environmental protection, a large number of technical means of eliminating the impact of pigments on people’s lives and the environment have been widely studied—for instance, turbidity point extraction [24,25,26,27], precipitation [28,29,30,31,32], membrane filtration [33,34], adsorption [35,36,37,38] and photocatalytic degradation [39,40,41,42]. Among them, adsorption has become the main treatment method for textile industry wastewater, with many advantages, such as low material price, little required equipment, a simple treatment process and superior treatment effects; thus, it is widely used. The key to this technology is the choice of adsorbent, such as activated carbon, which is of interest because it requires only a simple design, is easy to handle, and is sensitive to toxic substances, but its high cost has prevented its widespread use as an adsorbent in developing countries. Thus, low-cost natural materials, industrial wastes, and agricultural by-products have attracted the interest of researchers as alternatives to those high-cost adsorbents, such as rubber peels [43], sawdust [44], rice husks [45], and lignocellulose [46]. High purity mesoporous alumina (HPMA) with its large specific surface, suitable pore structure, excellent physical properties, chemical stability and activity has been extensively employed in the area of Congo Red adsorption. In combination with the structural characteristics of HPMA itself, the following aspects are considered to improve its adsorption performance: increasing the specific surface area of HPMA [23]; controlling the structural properties of alumina texture, i.e., the pore structure of alumina [47,48]; spherification of alumina powder morphology to improve its water permeability [49]; and the use of composites of other functional active materials to improve its adsorption activity [50,51]. The focus of this study is the synthesis of HPMA with excellent adsorption properties and high adsorption rates of Congo Red using lower-cost industrial aluminum hydroxide by direct ageing and the ammonium salt substitution method to increase the pore structure and specific surface area of the material, thus enhancing the surface energy and surface bonding energy, which can be easily stabilized by bonding with other atoms and enhance the chemical activity of HPMA. The greater chemical activity induces corresponding changes in atomic transport and conformation on the surface of the alumina nanoparticles, leading to changes in the electron spin conformation and electronic energy spectrum on the surface of the HPMA material particles, which contributes to the improved adsorption performance of HPMA. Moreover, the influence on purity, crystalline structure, porosity, morphology and the Congo Red dye adsorption properties of certain key preparation parameters are also investigated.

## 2. Experimental Procedures

### 2.1. Materials

Industrial Al(OH)_3_ was offered by an aluminum company in Guangxi, China, the main composition of which is shown in Table 2. The other analytically pure reagents used, hydrochloric acid, ammonium oxalate, ammonium acetate and ammonium carbonate, were all purchased from Xi-long Chemical Co., Shantou, China, and were used without any pretreatment.

### 2.2. Preparation of Mesoporous Alumina Materials

A mass of 219.4 g of NaOH was dissolved in 500 mL of ultra-pure water to obtain a highly-concentrated sodium hydroxide solution, 281.4 g of industrial Al(OH)_3_ was slowly added into the NaOH solution with magnetic stirring under hydrothermal conditions at 95 °C until all solids dissolved, and then it was filtered and diluted to obtain a range of sodium aluminate solutions with diverse initial concentrations of 30 g L^−1^, 60 g L^−1^, 90 g L^−1^ and 120 g L^−1^. The pH value of the NaAl(OH)_4_ solution was adjusted to 10 by slowly dropping in a dilute hydrochloric acid solution. The NaAl(OH)_4_ solution was aged at appropriate temperature (30 °C, 60 °C, 90 °C and 120 °C) for a period of time (1 h, 1.5 h, 2 h, 2.5 h, 3 h, 3.5 h and 4 h), and washed to neutral after centrifugation. The ammonium-rich γ-AlO(OH) was obtained by adding excess ammonium solution ((NH_4_)_2_CO_3_, CH_3_COONH_4_ and (NH_4_)_2_C_2_O_4_) for ammonium substitution reaction. The colloidal product was dried in a blast-drying oven at 150 °C for more than 2 h, and then roasted under a constant temperature (500 °C, 700 °C, 900 °C and 1100 °C) for 4 h with temperature rising at a rate of 10 °C min^−1^ to obtain the product, high purity mesoporous γ-Al_2_O_3_.

### 2.3. Characterization of Alumina Materials

The impurity content of alumina (iron, alumina, and silicon) was measured by X-ray fluorescence spectrum analysis (XRF, XRF-1800, Shimadzu Japan Ltd., Shanghai, China) to obtain the individual impurities removal rates from alumina products, and was calculated with the following formula:*R* = (*M*_0_ − *M_t_*)/*M*_0_


M_0_: the individual impurities mass ratio of industrial Al(OH)_3_ shown in Table 1.*M_t_*: the individual impurity’s mass ratio of alumina product.*R*: the individual impurity’s removal rate.

The morphology of the material was studied using a scanning electron microscope (SEM, S-4800, Hitachi Works, Ltd., Tokyo, Japan) worked at 5.0 KV and transmission electron microscopy (TEM, JEM-2100F, JEOL, Beijing, China). Structural phase analysis was carried out by X-ray diffraction (XRD) using CuKα radiation. The equipment was X’Pert PRO (PANalytical B.V., Almelo, The Netherlands), and the continuous mode was used to collect 2θ data from 10° to 80° with an 8° min^−1^ sampling pitch. The thermal stability of the adsorbent was determined using a Q500 thermogravimetric analyzer from TA company. The temperature range was 35–800 °C with a nitrogen (N_2_) atmosphere and the rate of increase was 10 °C min^−1^. The specific surface area, pore volume and pore size distribution of the sorbent were determined by liquid nitrogen adsorption using the surface area and pore porosimetry analyzer NoVA 1200e (Quantachrome Instruments, Shanghai, China) calculated using the Brunauer–Emmett–Teller (BET) and Barret–Joyner–Hallender (BJH) methods, respectively. The zeta (ζ) potential of the adsorbents was measured using a nanoparticle size and zeta point analyzer (Nani-ZS, Malvern, UK) at a pH value of 7.

### 2.4. Adsorption Experiments on HPMA Materials

The adsorption property of HPMA powder and its kinetic study were conducted. Two-hundred milliliters of Congo Red aqueous solution (mass concentration of 250 mg L^−1^) was added in a conical flask, adjusted to the pH value of 4 through 0.1M hydrochloric acid and 100 mg of the as-prepared HPMA material. The conical flask was then placed in a thermostatic water bath (DF-101S, Gongyi, China) with a controlled stirring speed of 145 rpm and a temperature of 25 °C. The solution was removed at pre-determined time intervals and the concentration of Congo Red in the removed solution was measured using Shimadzu UV-9000S spectrophotometer. The solution was then returned. The amount of sorption *q_t_* at different times was calculated using the following equation.
*q_t_* = (*C*_0_ − *C_t_*) *V*/*m*

C_0_: the initial concentration of Congo Red (mg L^−1^)*C_t_*: the mass content of Congo Red for the corresponding time (mg L^−1^)*V*: the volume of solution (L)*m*: mass of HPMA materials (g)*q_t_*: mass of adsorbent adsorbed per unit mass of adsorbent at time t

To study the adsorption isotherms, 100 mg of adsorbent was weighed and dispersed in 200-mL CR solutions with mass concentrations of 50–1750 mg L^−1^. These solutions were stirred for 24 h until adsorption equilibrium, and 4 mL of the supernatant after centrifugation was analyzed by UV-Vis spectrometer (UV-9000S, Shimadzu, Shanghai, China). The maximum adsorbed mass *q*_m_ for CR after equilibrium was calculated by the following equation.
*q_m_* = (*C*_0_ − *C_e_*) *V*/*m*

C_0_: the initial concentration of Congo Red (mg L^−1^)*C_e_*: the mass content of Congo Red at adsorption equilibrium (mg L^−1^)*V*: the volume of solution (L)*m*: mass of HPMA materials (g)*q_m_*: the maximum adsorbed mass

## 3. Results and Discussion

### 3.1. Effects of Various Factors on the Mesoporous Alumina Product

#### 3.1.1. Initial Concentration of NaAl(OH)_4_ Solution

The initial concentration of NaAl(OH)_4_ solution is a key parameter to the preparation of high-purity MA material. Therefore, a series of alumina materials were prepared with the ageing temperature of 60 °C, ageing time of 2 h, and the roasting temperature of 500 °C. The mass concentration of the (NH_4_)_2_CO_3_ solution was 75 g L^−1^ and for the NaAl(OH)_4_ solutions, with different mass concentrations, were 30 g L^−1^, 60 g L^−1^, 90 g L^−1^ and 120 g L^−1^. The solutions were prepared to study the effect of the initial mass concentration on the impurities contents of the products. As shown in Figure 1 and Table 3, with the increase of the initial concentration of NaAl(OH)_4_ from 30 g L^−1^ to 120 g L^−1^, the iron oxide content gradually increased from 0.0048% to 0.009%, the silica content initially decreased from 0.0098% to 0.0054% and then increased to 0.0083%; the sodium oxide content similarly firstly decreased from 0.046% to 0.022% and then increased to 0.038%. The overall impurity removal rate, on the other hand, increased from 97.97% to 98.86% and then decreased to 98.15% and the purity of alumina rises from 99.9393% to 99.9661% and then falls to 99.9447%.
Al(OH)_3_ + NaOH = NaAl(OH)_4_;
Fe_2_O_3_ + 2NaOH + 3H_2_O = 2NaFe(OH)_4_;
SiO_2_ + 2NaOH = Na_2_SiO_3_ + H_2_O;
2NaAl(OH)_4_ + 1.7Na_2_SiO_3_ = Na_2_O Al_2_O_3_·1.7SiO_2_·*n*H_2_O↓ + 3.4NaOH + 3H_2_O

From the above reaction equation, it can be seen that the SiO_2_ impurities present in the raw material reacted with NaOH to form NaSiO_3_, and NaSiO_3_ further reacted with NaAl(OH)_4_ to form precipitates, which facilitated the removal of SiO_2_. On the other hand, iron oxide reacted with sodium aluminate to form soluble NaFe(OH)_4_, which increased the amount of iron oxide impurities in the MA. Meanwhile, as the initial concentration gradually increased, the concentration of sodium ions rose and the viscosity of the solution increased, so that the content of each impurity in the product showed an increasing trend after the concentration exceeded 60 g L^−1^.

#### 3.1.2. Aging Temperature

As with other process parameters, aging temperature had a definite effect on the purity of the final product. A series of alumina materials were prepared with a NaAl(OH)_4_ solution concentration of 60 g L^−1^, an ageing time of 2 h, the roasting temperature of 500 °C, the mass concentration of the (NH_4_)_2_CO_3_ solution of 75 g L^−1^ and the ageing temperatures 30 °C, 60 °C, 90 °C and 120 °C. The results of the effect of different aging temperature on the impurity content in the product alumina are shown in Table 4 and Figure 2. It can be seen that with the increase of aging temperature from 30 °C to 60 °C, there was no significant change in the mass fractions of iron oxide or silica, while the content of sodium oxide decreased significantly (from 0.04% to 0.022%). When the temperature exceeded 60 °C, the mass ratio of all impurities increased, from 0.0065% to 0.0081% for Fe_2_O_3_, from 0.0054% to 0.0067% for silica, and to 0.030% for Na_2_O. The total removal of impurities was first increased, from 98.26% to 98.86%, and then decreased to 98.53%, and the purity of alumina rose from 99.9481% to 99.9661% and then fell to 99.9552%. Excessive increases in ageing temperature, therefore, not only result in a loss of energy but also reduce the purity of the alumina. The reason for this change is that when the temperature was too low (less than 60 °C), the viscosity of the slurry was too large for the removal of impurities and the final precipitation produced resulted in an increase in the amount of sodium impurities contained in the material. When the temperature was raised to 60 °C, the solution had a more suitable viscosity, there were more nuclei, the Al(OH)_3_ particles formed were smaller, the crystals had less effect on the impurities, and fewer impurities remained and were adsorbed. However, as the temperature gradually rose, the crystals formed were too large, the force between the crystals was enhanced, the viscosity of the solution was greater at this time, and the adhesion effect on the impurities was greater. So, after the temperature exceeds 60 °C, the content of all the impurities increased, and the effect of impurity removal gradually became worse.

#### 3.1.3. Aging Time

Ageing time is also an important parameter in the synthesis and preparation process, so a careful study of ageing time is of general importance. A collection of alumina materials were synthesized with a NaAl(OH)_4_ solution concentration of 60 g L^−1^, ageing temperature of 60 °C, roasting temperature of 500 °C, a mass concentration of the (NH_4_)_2_CO_3_ solution of 75 g L^−1^ and the ageing times 1 h, 1.5 h, 2 h, 2.5 h, 3 h, 3.5 h, 4 h. The effects of ageing time on the purity and impurity removal rate of the material are shown in Figure 3 and Table 5, respectively. When the time was extended from 1 h to 2 h, there was no significant change in the content of iron oxide and silica, while the content of sodium oxide decreased significantly (from 0.038% to 0.022%). When the time exceeded 2 h (2.5–4.0 h), the content of all impurities increased, from 0.0065% to 0.0079% for Fe_2_O_3_, from 0.0054% to 0.0071%, for silica and to 0.048% for sodium oxide. The total removal of impurities first increased from 98.31% to 98.86% and then decreased to 97.90%, and the purity of the alumina rose from 99.9497% to 99.9661% and then fell to 99.9371%, which indicated that the aging time had a greater impact on the impurity removal effect. When the aging temperature was low, the solution concentration viscosity was larger and the adsorption force of aluminum hydroxide crystals on impurities was larger, lowering the impurity removal rate, so extending the aging time is conducive to the removal of impurities. When the time was extended to more than 2 h, the crystals slowly grew and agglomerated, and the inter-crystalline and crystal forces increased, which is not conducive to the removal of impurities. Thus, when the ageing time is 2 h, the impurity Na_2_O content in the product alumina was lowest.

#### 3.1.4. Type of Ammonium Salt

To investigate the effect of ammonium salt type on the alumina powder quality, three alumina materials were prepared with a NaAl(OH)_4_ solution concentration of 60 g L^−1^, an ageing temperature of 60 °C, an ageing time of 2 h, a roasting temperature of 500 °C, the mass concentrations of the ammonium oxalate, ammonium acetate and ammonium carbonate solutions of 75 g L^−1^. The nitrogen adsorption–desorption isotherms and pore-size distribution plots of alumina materials washed with diverse ammonium salt solutions are shown in Figure 4; it shows that all the alumina particles showed the classical shape of a Type IV isotherm curve with an H1-type hysteresis loop, and all samples are known to be mesoporous materials, according to the IUPAC classification [52]. The effects of ammonium oxalate, ammonium acetate and ammonium carbonate on the impurity mass fraction and specific surface area of the product alumina are shown in Table 6; it shows that the type of ammonium salt had no obvious effect on the Fe_2_O_3_ and SiO_2_ mass ratio in the alumina product, but the type of ammonium salt had a large effect on the content and porosities of sodium impurities in the product. The sodium content of the alumina material prepared using ammonium carbonate as a sodium removal agent was only 0.022%, with a specific surface area of 312.43 m^2^ g^−1^ and a pore volume of 0.43 cm^3^ g^−1^, whereas, the sodium impurity content using ammonium acetate and ammonium oxalate was as high as 0.054% and 0.050% respectively, with specific surface areas of 276.85 m^2^ g^−1^ and 252.20 m^2^ g^−1^, respectively, with pore volumes of 0.32 cm^3^ g^−1^ and 0.33 cm^3^ g^−1^, also respectively. This is due to the fact that ammonium carbonate can release more NH_4_^+^ in the hydrolysis process, and NH_4_^+^ can replace sodium ions in the reaction process and be absorbed by γ-AlO(OH) [53,54]. After roasting to until the release of ammonia, it makes the solution more effective in removing sodium ions and achieves the best sodium removal. Gas release increased the specific surface area of the product and its porosity. On balance, the choice of ammonium carbonate washing is more appropriate.

#### 3.1.5. Mass Concentration of (NH_4_)_2_CO_3_ Solution

From the previous description, it is clear that ammonium carbonate had a good contribution to the purity and porosity of the material, so it is important to investigate the effect of different ammonium carbonate mass concentrations on the purity and porosity of the product. A collection of alumina materials was synthesized with a NaAl(OH)_4_ solution concentration of 60 g L^−1^, an ageing temperature of 60 °C, an ageing time of 2 h, a roasting temperature of 500 °C and the mass concentrations of the (NH_4_)_2_CO_3_ solutions of 25 g L^−1^, 50 g L^−1^, 75 g L^−1^, 100 g L^−1^. As shown in Table 7 and Figure 5, with the increase of the mass concentration of ammonium carbonate from 25 g L^−1^ to 75 g L^−1^, the sodium oxide content decreased significantly (from 0.039% to 0.022%), the sodium oxide removal rate increased (from 93.28% to 96.21%) and the purity of alumina rose from 99.9478% to 99.9661% and then fell to 99.9653%. When the mass concentration was increased to 100 g L^−1^, the sodium oxide content and removal rate remained essentially unchanged. Changing the mass concentration of ammonium carbonate did not affect the impurity contents of iron oxide or silica, nor their removal rates. At low concentrations, as the concentration increased (from 25 g L^−1^ to 75 g L^−1^), there was a gradual increase in the number of ammonium ions hydrolyzed from the solution, and, therefore, more sodium impurities were replaced by ammonium ions, which improved the purity of the material and also increased the specific surface area of the material. However, as the concentration became too high, the number of sodium impurities that could be replaced became a certain amount, so continuing to increase the concentration of ammonium carbonate later has no effect on the quality of the product. In summary, the HPMA powder synthesized by washing with 75 g L^−1^ ammonium carbonate solution had the highest purity and the largest specific surface area in terms of cost savings.

#### 3.1.6. Roasting Temperature

To investigate the effect of firing temperature on the structure and properties of the products, four parts of NaAl(OH)_4_ solution were prepared, hydrochloric acid solution was added. The solution was aged at 60 °C for 2 h and filtered, washed with ammonium carbonate solution (mass concentration of 75 g L^−1^), dried and roasted at 500 °C, 700 °C, 900 °C and 1100 °C to obtain the product alumina. It can be seen from the XRD patterns (Figure 6) that the product is γ-Al_2_O_3_ with no significant change in the crystalline phase when roasted at 500–700 °C; this partly occurs in the product-phase transformation from γ-Al_2_O_3_ to the θ-Al_2_O_3_ crystalline phase when roasted at 900 °C, and to the θ-Al_2_O_3_ crystalline phase, which has a small specific surface area (JCPDF Card: No. 11-0517) and is completely formed at 1100 °C. θ-Al_2_O_3_ is the most stable tripartite crystalline system among all alumina phases; it has a low specific surface area and weak chemical activity, so it is not suitable for the preparation of adsorbents or catalysts. Therefore, it is not suitable for the preparation of adsorbents or catalysts.

The nitrogen adsorption–desorption isotherms and the pore size distribution plots of alumina materials calcined for 4 h at various temperatures are shown in Figure 6b,c. The alumina particles roasted at various temperatures show the classical shape of a Type IV isotherm curve with an H1-type hysteresis loop, and all samples are known to be mesoporous materials according to the IUPAC classification [52]. It can be revealed from Table 8 that an increase in calcination temperature from 500 °C to 1100 °C caused the pore size of the alumina material to increase and the number of pores to decrease. This is because under high temperature conditions, the entire pore structure on the surface of the material collapses and agglomerates, as shown in TEM images (Figure 7), resulting in a dramatic reduction in specific surface area and total pore volume (the specific surface area was reduced from 312.43 m^2^ g^−1^ to 88.52 m^2^ g^−1^, pore volume 0.48 cm^3^ g^−1^ to 0.08 cm^3^ g^−1^.) Thus, the HPMA material prepared by roasting at 500 °C had the largest specific surface area and pore volume, with more active chemistry sites for catalysts and adsorbents.

### 3.2. Characteristics of High Purity Mesoporous Alumina

Gamma-phase high-purity mesoporous alumina (HPMA) materials with high specific surface areas were successfully prepared by the direct ageing and ammonium salt replacement methods. As the sodium hydroxide solution dissolved the industrial aluminum hydroxide, the Si and Fe impurities could be effectively removed by dissolving the industrial aluminum hydroxide with sodium hydroxide which has been described in Section 3.1.1. The solution equilibrium was disrupted by the direct ageing of dilute hydrochloric acid drops into the reaction system. γ-AlO(OH) crystals are rapidly generated with fine crystal particles, resulting in weak intermolecular forces; therefore, the adsorption of impurities was weak, while the fine particles facilitated an increase in the specific surface area of the material. Finally, by adding an excess of ammonium salt solution, the dissociated NH_4_^+^ could replace the sodium ions during the reaction to produce Na_2_CO_3_ [55]. NH_4_^+^ can be absorbed by γ-AlO(OH) [53,54], which plays a significant desaturation role, and ammonium ions on the surface of γ-AlO(OH) were released in the form of ammonia molecules during calcination, increasing the porosity of the material and thus improving the specific surface area. The main chemical reactions in the preparation process are as follows:Al(OH)_3_ + NaOH→NaAl(OH)_4_;
Fe_2_O_3_ + 2NaOH + 3H_2_O→2NaFe(OH)_4_;
SiO_2_ + 2NaOH→Na_2_SiO_3_ + H_2_O;
2NaAl(OH)_4_ + 1.7Na_2_SiO_3_→Na_2_O·Al_2_O_3_·1.7SiO_2_·nH_2_O↓ + 3.4NaOH + 3H_2_O; 
NaAl(OH)_4_ + HCl→Al(OH)_4_^−^ + NaCl + H^+^;
NaAlO_2_→NaOH + Al(OH)_4_^−^;
NaOH + (NH_4_)_2_CO_3_→Na_2_CO_3_ + NH_3_·H_2_O;
(NH_4_)_2_CO_3_→2NH_4_^+^ +CO_3_^2−^;
Al(OH)_4_^−^→γ-AlO(OH)·nH_2_O + (1 − n)H_2_O;
2γ-AlO(OH)·nH_2_O→γ-Al_2_O_3_ + 2(1 − n)H_2_O;

The impurity mass ratios, excellent removal rates and surface physical properties of the HPMA materials synthesized under optimum conditions with initial NaAl(OH)_4_ solution concentration of 60 g L^−1^, ageing temperature of 60 °C, ageing time of 2 h, roasting temperature of 500 °C and the mass concentration of the (NH_4_)_2_CO_3_ solution of 75 g L^−1^ are shown in Table 9 revealing that the mass fractions of residual Fe_2_O_3_, SiO_2_ and Na_2_O in the alumina products were 0.0065%, 0.0054% and 0.022% respectively, with removal rates of 99.44%, 99.57% and 96.21% respectively, which showed that the purity of the product was as great as 99.9661% and the total removal rate of impurities was 98.87%. Moreover, the specific surface area, pore volume and pore size of the material were 312.43 m^2^ g^−1^, 0.48 cm^3^ g^−1^ and 3.80 nm, respectively, indicating that the material prepared was HPMA with high purity and porosity, which can be widely used in the preparation of adsorbents.

XRD patterns (Figure 8a) showed that the diffraction peaks at 2θ = 19.439°, 31.951°, 37.648°, 39.398°, 45.854°, 60.878° and 67.029° correspond to the (111), (220), (311), (222), (400), (511) and (440) crystals of γ-Al_2_O_3_ (JCPDS card NO. 10-0425), respectively. There was not any other significant diffraction peak, indicating that this alumina with a cubic structure had the absolute high purity [56]. All diffraction peaks exhibited a high degree of broadness for their fine natures, and less degeneracy in the crystal. According to the Debye–Scherrer equation [57], D = k λ/β cosθ, where k is a constant, θ is the diffraction angle, λ is the X-ray wavelength and β is the full width at half maximum (FWHM), the principal grain sizes of γ-Al_2_O_3_ calculated from the full width at half maximum of the isolated (311), (400) and (440) diffraction peaks are 14.3 nm, 12.8 nm and 16.7 nm, respectively.

Figure 8b shows the TGA diagram of the adsorbent HPMA material. As can be seen from the graph, there was no distinctive weight loss with the temperature rising from room temperature to 800 °C, proving that the mesoporous material produced has good thermal stability, which can be applied in the adsorption of Congo Red.

It can be seen from the SEM picture (Figure 8c) and TEM image (Figure 8d) that the product alumina was porous, small particles with clear individual particles, less agglomeration and many fine pore channels on the surface of the product, which increased the specific surface area of the material, enhancing the surface energy and surface bonding energy, which can be easily stabilized by bonding with other atoms, thus increasing the chemical activity of HPMA. The greater chemical activity induced corresponding changes in atomic transport and conformation on the surface of the alumina nanoparticles, leading to changes in the electron spin conformation and electronic energy spectrum on the surface of the HPMA material particles, features that contribute to the improved adsorption performance of HPMA. The size of the crystals was essentially in the range of 200 nm to 500 nm. Zeta potential is an important parameter in measuring the strength of mutual repulsion or attraction between particles; as the absolute value of the zeta potential rises, the degree of dispersion is much greater than the degree of aggregation, making the molecular surface attraction greater than the repulsion. The zeta potential plots of the as-synthesized HMPA nano-materials are shown in Figure 8e, showing that the zeta potential value of the HPMA particles was 46 ± 3 mV, with potential values in the range of 40–60 mV, revealing that the adsorbent had a positive charge on its surface with outstanding surface stability [58] and excellent adsorption property for CR.

Nitrogen adsorption–desorption isotherms are shown in Figure 8f; according to the IPUAC classification [52], the HPMA powder shows a typical type IV isotherm curve shape with an H1-type hysteresis loop, which is due to the slight sintering of mesoporous γ-Al_2_O_3_ at this calcination temperature. The pore size distribution curve, cumulative pore volume plot and cumulative surface area curves of the mesoporous γ-Al_2_O_3_ material obtained by the BJH model are shown in the inset of Figure 8f–h, respectively. As can be seen from the pore size distribution plots, the pore size of the material was mainly distributed in the range of 2–16 nm, while the cumulative pore volume curves and cumulative surface area curves of the material were in the range of 2–24.82 nm. Mesoporous materials are defined as pore sizes in the range of 2–50 nm porous material [59], therefore, the product was a porous, highly dispersed solid material with a large specific surface area and high activity, which confer good catalytic and adsorption properties [1,2,3,4,5].

### 3.3. HPMA Materials Adsorption of Congo Red Analysis

Adsorption kinetics studies play a critical role in adsorption studies, as they can predict very accurately the rate at which pollutants are removed during the adsorption process. Additionally, adsorption kinetic studies can also provide quite pragmatic data when speculating on adsorption mechanisms. The kinetic curve for the adsorption of Congo Red dye in the HPMA sample is shown in Figure 9a. The adsorption process was carried out at 25 °C, the concentration of Congo Red dye was fixed at 250 mg L^−1^ and the amount of adsorbent was 100 mg. It shows that at the beginning of the process (from 0–10 min), adsorption proceeded very rapidly, but after saturation of the active centers from the adsorbent surface, adsorption remained unchanged. This is due to the fact that the remained vacant sites were difficult to occupy, due to repulsive forces between the dye adsorbed on the surface of HPMA and the solution phase [60,61]. The adsorption equilibrium was reached in about 60 min, the adsorption amount was 492.19 mg g^−1^ at that time, and the dye removal rate was as high as 98.44%. As discussed in the nitrogen adsorption–desorption analysis, a desodium agent can enahnce the pore structure of the material to increase the specific surface area of the material, and thus enhance its chemistry activity, increasing adsorption capacity. The adsorption kinetic data are in good agreement with this speculation. It is suggested that this HPMA material has excellent adsorption performance for CR and the adsorption rate is very fast, thus, it is suitable for wide use in the adsorption of CR in organic dyes.

Having analyzed the kinetic characteristics of the adsorbent HPMA for Congo Red dye, it is possible, in practice, to design the Congo Red adsorption process in solution. Typically, the frequently used kinetic models are as follows:

Pseudo-first-order model:ln(*q_e_* − *q_t_*) = ln*q_e_* − K_1_t

Pseudo-second-order model:t/*q_t_* = 1/K_2_*q^2^_e_* + t/*q_e_*
in which *q_e_* and *q_t_* (mg g^−1^) are, respectively, the adsorption capacities at equilibrium and at any time t (min).

The kinetic behavior of Congo Red adsorption on the synthesized HPMA was investigated using pseudo-first-order and pseudo-second-order kinetic models. The obtained kinetic parameters for adsorption are listed in Table 10 and these fitted curves are shown in the Figure 9b,c. Apparently, when using the pseudo-first-order kinetic model, the experimental value (*q_e_*) was much higher than the calculated value, probably due to the fact that the correlation coefficient, *R*^2^, was as low as 0.59214. Contrarily, the pseudo-second-order kinetic model was used to obtain the calculated value of 495.05, close to the experimental value of 495, and the correlation coefficient *R*^2^ was as high as 0.99992, indicating that the adsorption process of Congo Red on synthetic HPMA materials is very consistent with the pseudo-second-order kinetic model.

In order to further investigate the internal particle diffusion, the experimental data were fitted with the internal particle diffusion model proposed by Weber and Morris [62]. For adsorbents, the distribution of the adsorbed amount with respect to time t^0.5^ is more regular than the distribution with respect to contact time t. This model was obtained by fitting the following equation.
*q_t_* = K_di_ t^0.5^ + C_i_

where K_di_ is the rate constant for stage i, calculated from the slope of the curve of *q_t_* with respect to t^0.5^. C_i_ is the intercept of stage i; the larger the intercept, the more pronounced the boundary layer effect will be. For the internal particle diffusion model, *q_t_* should be linear with respect to t^0.5^, and if the curve passes through the origin point, the velocity-limiting process only depends on the internal particle diffusion rate. A linear fit of the internal particle diffusion model for the adsorption of Congo Red was shown in Figure 9d, where it can be seen that the whole adsorption process can be divided into three steps. From this model, transient adsorption, which can also be referred to as external adsorption, occurs in the solution during the first 2 min when the rate of adsorption is high due to the high initial concentration of Congo Red. The rate of adsorption then decreases significantly during the second phase, which is known as the progressive or slow adsorption phase, and it is the stage wherein the rate of adsorption is limited. The last one is the final equilibrium stage, wherein the CR concentration is at its lowest and therefore the internal particle diffusion rate is at its slowest. The phenomenon K_d1_ < K_d2_ < K_d3_ was shown in Table 11.

The adsorption isotherm is a mathematical model used to describe the adsorbed species distribution on the adsorbent surface of the adsorbent [63,64]. The nonlinear isotherm models proposed by Langmuir [65], Freundlich [66] and Temkin [67] were used to investigate the adsorption mechanism. These models were chosen to understand whether adsorption occurs in monolayers (Langmuir), multilayers (Freundlich) or by diffusion (Temkin).

Langmuir model:*q_e_* = *q_m_ K_L_ C_e_*/(1 + *K_L_ C_e_*)
where *C_e_* (mg/L) is the equilibrium concentration of solute, *q_e_* (mg/g) is the equilibrium adsorption capacity of the adsorbent, *q_m_* (mg/g) is the saturated adsorption amount of the adsorbent and *K_L_* (L/mg) is the Langmuir adsorption constant.

Freundlich model:*lnq_e_* = *lnK_F_* + (1/*n*) *lnC_e_*

where *K_F_* (L mg^−1^) and *n* are Freundlich isotherm constants, which refer to the capacity and intensity of the adsorption.

Temkin model:*q_e_* = *RT lnK_T_*/*b* + *RT lnC_e_*/*b*
where *K_T_* (L mg^−1^) and *b* are Temkin isotherm constants, *R* (J/(mol K)) is the ideal gas constant, *T* (K) is the adsorption temperature.

To study the adsorption isotherms, 100 mg of adsorbent was weighed and dispersed in 200-mL CR solutions with mass concentrations ranging from 50–1750 mg L^−1^. The relation between the adsorption capacities, removal efficiency of the adsorbents and the initial concentrations of CR was given by the adsorption isotherm in Figure 10. It could be observed that the adsorption capacities of the high-porosity HPMA nanoparticles increased with increase in the initial CR mass concentration and then remained at 1984.64 mg g^−1^, while its removal efficiency decreased. The linear plots of the Langmuir model, Freundlich model and Temkin model of the adsorption of CR are given in Figure 10b–d and the adsorption isotherms parameters for adsorption of CR on the HPMA powder are shown in Table 12. It shows that the value of *R*^2^ of the Langmuir model for the prepared high-porosity HPMA material is 0.99972, while the *R*^2^ values of Freundlich model and Temkin model for this material were 0.86148 and 0.97039, respectively. Therefore, the adsorptions of the prepared high porosity HPMA material followed the Langmuir isotherm model closely. which indicates that the adsorption between the sample and Congo Red was a physical adsorption of a single molecular layer on its surface and that no chemical reaction took place [5]. Furthermore, the linearization fitted with the Langmuir isotherm model of the equations for the prepared material was y = 0.000486898x + 0.01022, and the *q_m_* value for the adsorption of CR by the prepared high porosity HPMA material calculated with Langmuir isotherm model was 2053.818 mg g^−1^, which was a little higher than the experiment data of 1984.64 mg g^−1^.

To investigate the effect of different pH on the saturation adsorption capacity, adsorption was carried out in solutions of pH 2, 4, 6, 8, 10 and 12 (Congo Red mass concentration of 250 mg L, adsorbent mass of 100 mg, temperature of 25 °C). The variation of adsorption and removal rates of Congo Red, with pH values of the solution increasing from 2 to 12, are shown in Figure 11. It can be seen that the equilibrium adsorption capacities of the as-synthesized HPMA nanoparticle ranged from 491.86 mg g^−1^ to 492.19 mg g^−1^ at pH values from 2 to 4. The saturation adsorption of Congo Red gradually decreased from 492.19 mg g^−1^ (pH value of 4) to 103.67 mg g^−1^ (pH value of 12) as the pH of the solution exceeded 4 (from pH value of 6 to of 12), and removal efficiency reduced from 98.44% (pH value of 4) to 20.73% (pH value of 12). The pH effects on adsorption capacity for Congo Red could be explained by the charge attraction; in the acid solution, the pyridines rings on the HPMA surface were promoted and attracted anionic dye, Congo Red. As the hydroxide ions in the solution increased with increasing pH, and as Congo Red ions and hydroxide ions carry the same electrical charge, the hydroxide ions also occupied adsorption sites on the surface of the material, which increased the competition between the hydroxide ions and the Congo Red ions [28,37,38,42], therefore, the adsorption capacity of Congo Red on the high-porosity HPMA material in the acid solution Was better than that in the base solution.

In order to evaluate the adsorption capacity of the HPMA adsorbents for the removal of CR dyes, and its adsorption maxima for Congo Red were compared with other as-reported metal hydroxides. Moreover, the comparison of the maximum monolayer adsorption (*q*_m_) for CR removal by various series of adsorbents is shown in Table 13, Further observation indicates that the maximum adsorption capacity of the as-prepared high-porosity HPMA nanoparticles for CR was significantly higher than that of the majority of reported materials. For instance, the adsorption capacity of the magnetic core-manganese oxide shell for CR was just 43.00 mg g^−1^, while the adsorption capacity of the as-synthesized alumina powder, with a high-porosity structure, for CR was 1984.64 mg g^−1^, about 46.15 times higher than that of the magnetic core-manganese oxide shell. The high adsorption capacity is due to the higher specific area and novel high-porosity nanostructures. It indicates that the HPMA material prepared in this study has excellent adsorption properties for Congo Red, and, thus, has the potential for wide application in the treatment of industrial dyes containing Congo Red.

## 4. Conclusions

In summary, we have synthesized HPMA with a porous structure using non-toxic, cheap sodium hydroxide and industrial aluminum hydroxide by a simple direct ageing method and ammonium salt substitution. The results show that HPMA materials have excellent thermal stability at the purity of 99.9661% and total impurities removal rates of 98.87%, and shows the advantages of both their microstructures and nanostructures, which avoid the aggregation of HPMA nanoparticles and maintain high specific surface areas, a high value of pore volume and ideal pore size distribution, enhancing the surface energy and surface bonding energy, which can be easily stabilized by bonding with other atoms to enhance the chemical activity of HPMA. The greater chemical activity induces corresponding changes in atomic transport and conformation on the surface of the alumina nanoparticles, leading to changes in the electron spin conformation and electronic energy spectrum on the surface of the HPMA material particles, contributing to the improved adsorption performance of HPMA. Moreover, the adsorption mechanism of CR in solution is electrostatic adsorption and the zeta potential of HPMA is 46 ± 3 mV, with potential values between 40–60 mV, indicating that the adsorbent has a positive surface charge and outstanding surface stability, conferring good adsorption performance the HPMA particle for CR. The maximum capacities of the HPMA nanostructures with high porosity for CR have been determined to be 1984.64 mg g^−1^, demonstrating its promising potential in environmental remediation. The underlying adsorption kinetics follow the pseudo-second-order model, and the adsorption isotherms follow the Langmuir model. The synthesis method is simple, versatile and controllable, which makes it possible to easily achieve efficient HPMA production at the gram level for the treatment of toxic dyes in wastewater. Owing to their highly porous structures, special surface properties and high surface areas, HPMA nanoparticles are potentially applicable in water purification.

## Figures and Tables

**Figure 1 materials-15-00970-f001:**
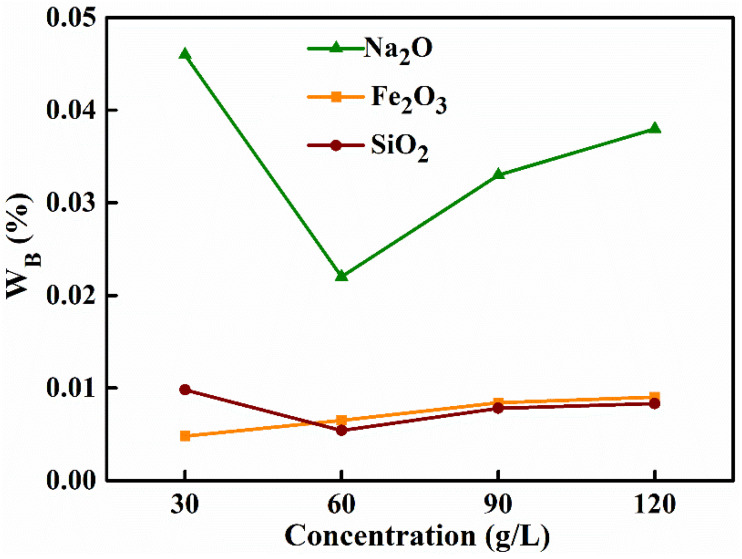
Effect of initial concentration of NaAl(OH)_4_ solution on the mass ratio of MA.

**Figure 2 materials-15-00970-f002:**
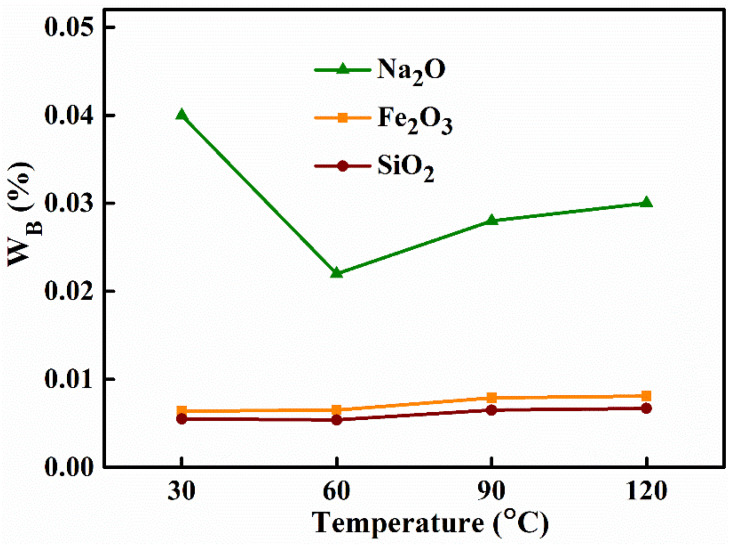
Effect of aging temperature on the impurity mass ratio in the product alumina.

**Figure 3 materials-15-00970-f003:**
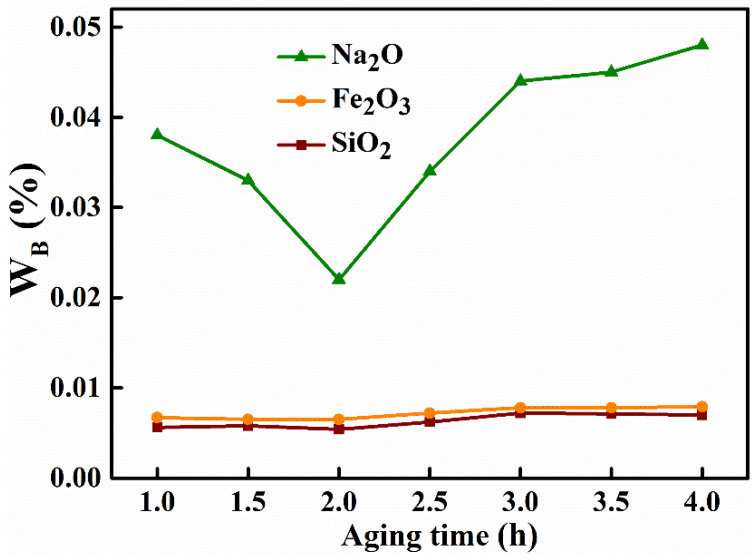
Effect of aging time on the impurity mass fraction in the product alumina.

**Figure 4 materials-15-00970-f004:**
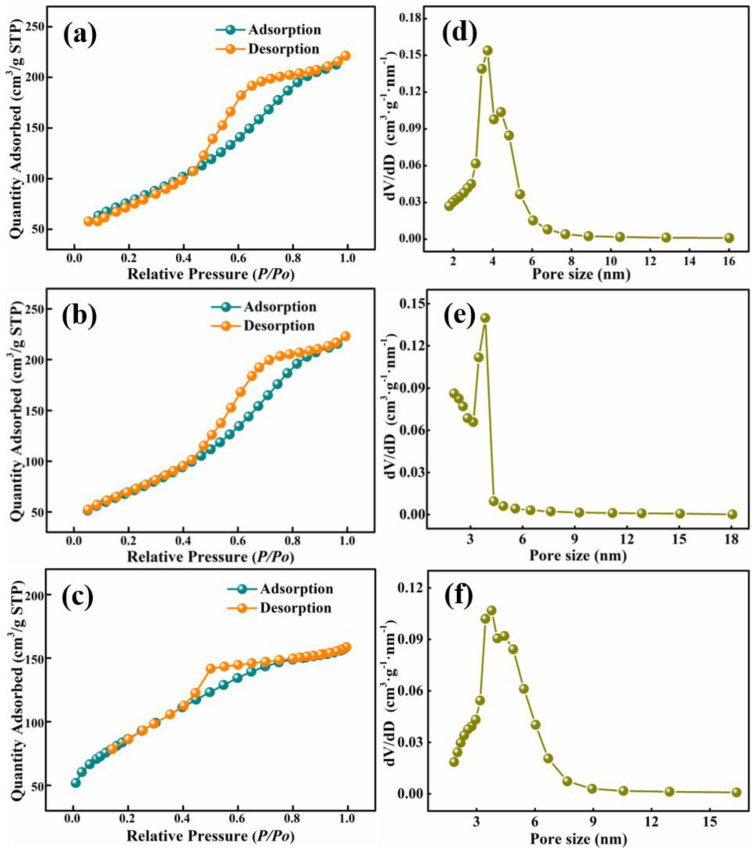
Nitrogen adsorption and desorption isotherms (**a**): (NH_4_)_2_CO_3_, (**b**): CH_3_COONH_4_, (**c**): (NH_4_)_2_C_2_O_4;_ pore size distribution curves of different ammonium salts used in the products (**d**): (NH_4_)_2_CO_3_, (**e**): CH_3_COONH_4_, (**f**): (NH_4_)_2_C_2_O_4_.

**Figure 5 materials-15-00970-f005:**
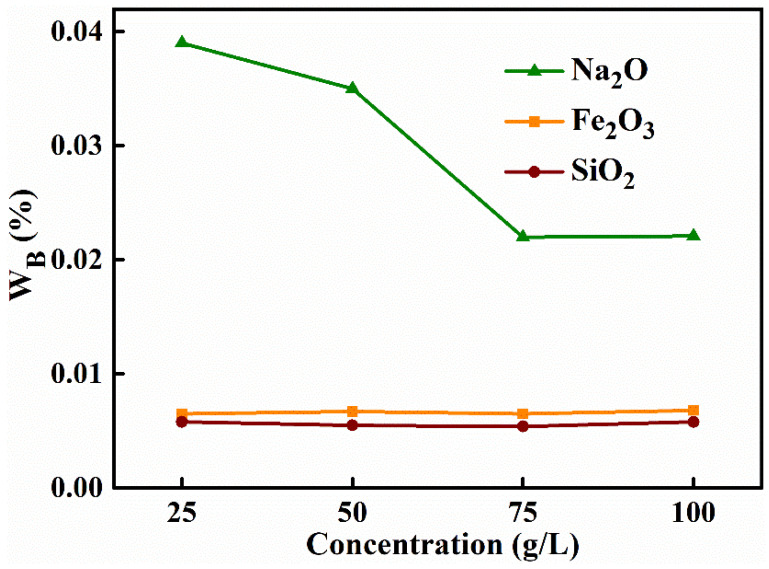
Effect of the mass concentration of ammonium carbonate solution on the mass fraction of the product alumina.

**Figure 6 materials-15-00970-f006:**
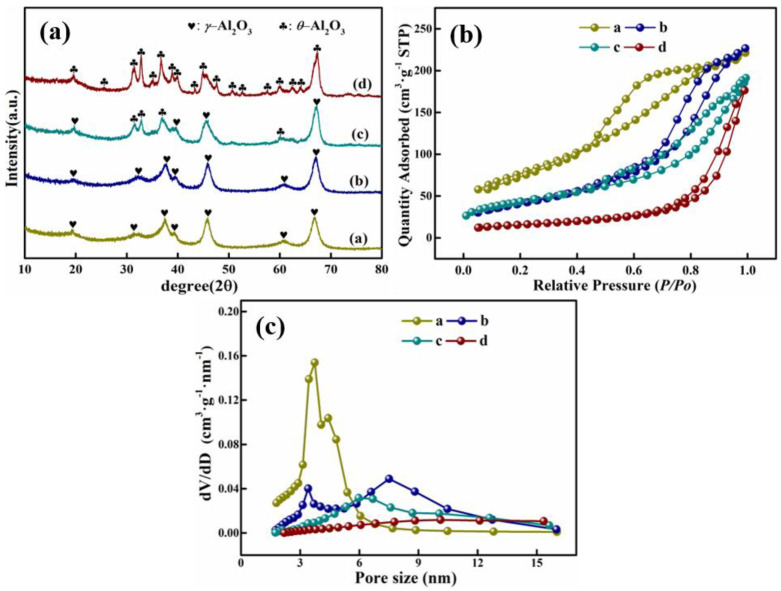
XRD patterns; (**a**) nitrogen adsorption-desorption isotherms; (**b**) pore-size distribution curves (**c**) of HPMA products obtained at different roasting temperatures: (**a**) 500 °C; (**b**) 700 °C; (**c**) 900 °C.

**Figure 7 materials-15-00970-f007:**
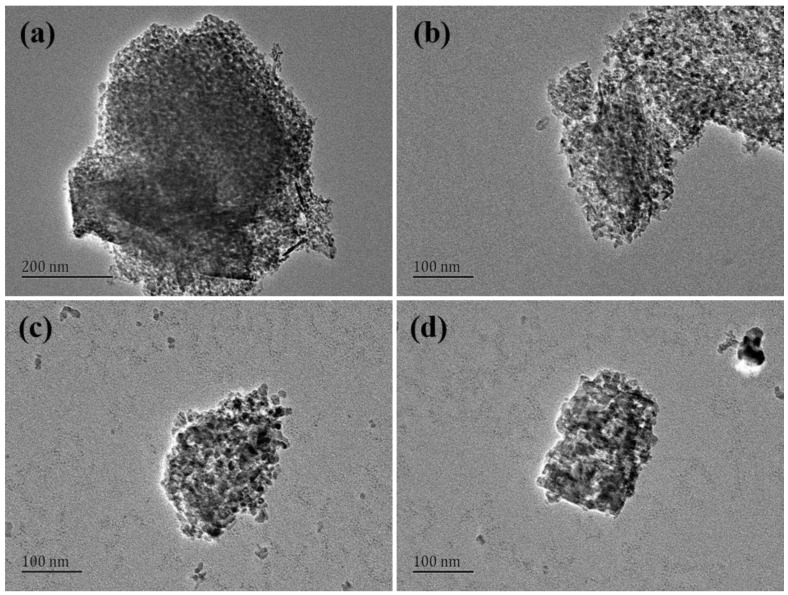
TEM images of HPMA materials obtained at different roasting temperatures: (**a**) 500 °C; (**b**) 700 °C; (**c**) 900 °C; (**d**) 1100 °C.

**Figure 8 materials-15-00970-f008:**
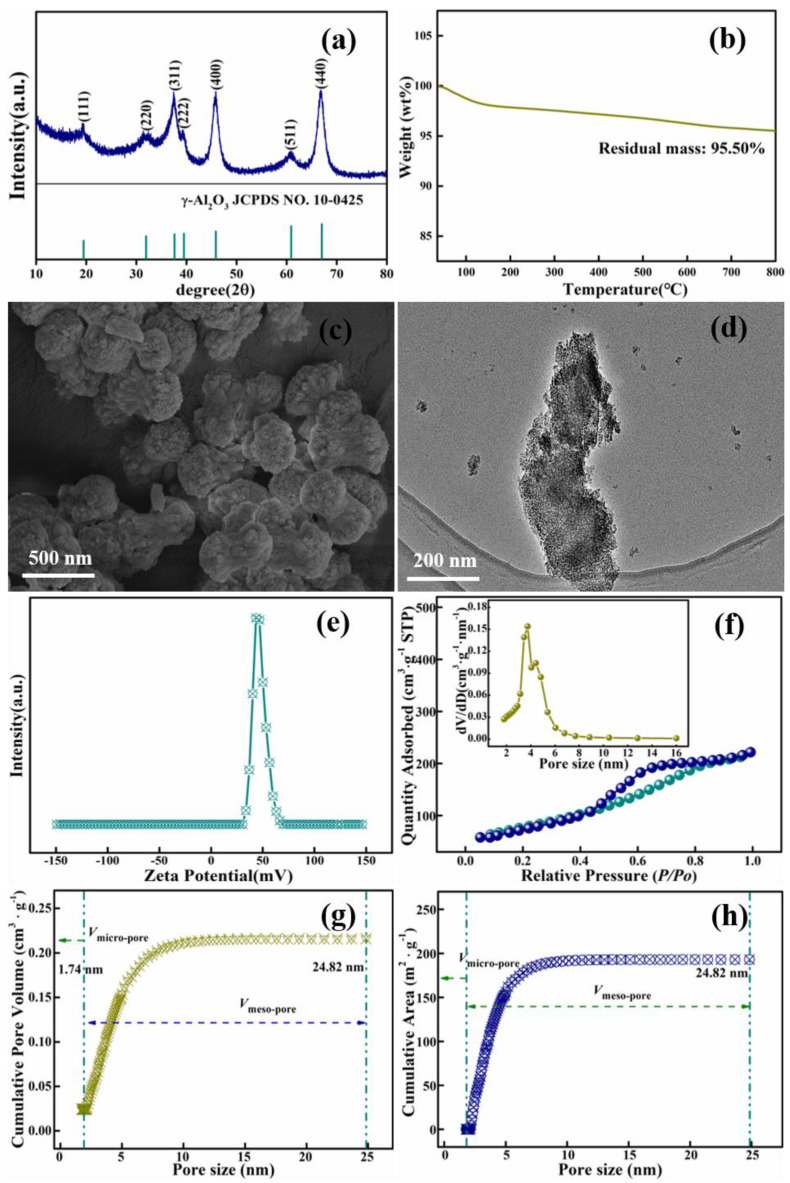
XRD pattern (**a**); TGA diagram (**b**); SEM image (**c**); TEM picture (**d**); Zeta potential plots (**e**); Nitrogen adsorption–desorption isotherms (**f**); cumulative pore volume (**g**); and cumulative area (**h**) through BJH kernels of mesoporous HPA material synthesized under optimal conditions. The inset of (**f**) shows the pore-size distribution curves in the BJH model.

**Figure 9 materials-15-00970-f009:**
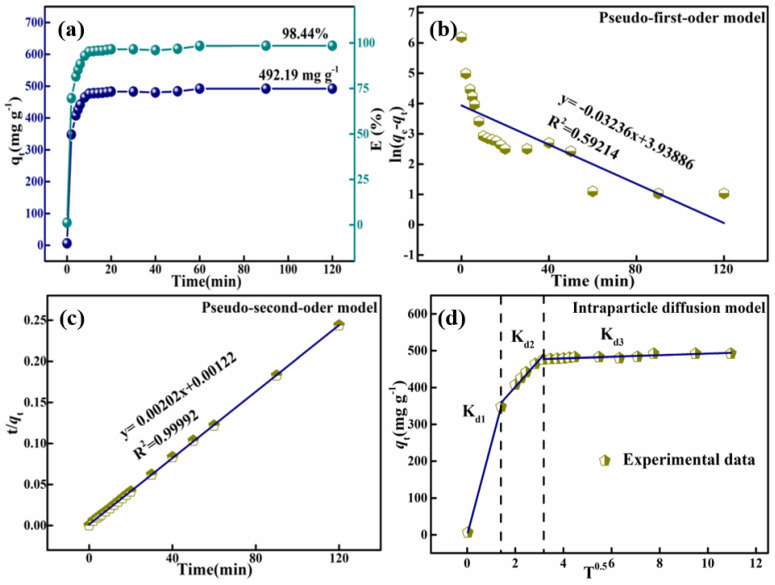
Variation in adsorption capacity with adsorption time for CR (**a**); pseudo-first order kinetics (**b**); pseudo-second order kinetics (**c**); intra-particle diffusion kinetics (**d**) for adsorption of CR on samples HPMA (T = 25 °C, adsorbent mass = 100 mg, CR concentration = 250 mg L^−1^, and pH = 4).

**Figure 10 materials-15-00970-f010:**
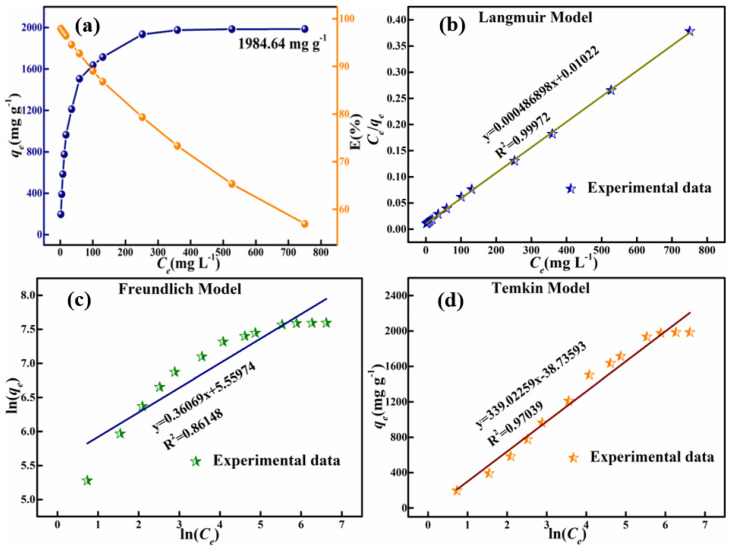
(**a**) Adsorption isotherms and percentage removal of CR as a function of the initial concentration; plots fitted with (**b**) Langmuir Model, (**c**) Freundlich Model and (**d**) Temkin Model for adsorption of CR on samples HPMA (T = 25 °C, adsorbent mass = 100 mg and pH = 4).

**Figure 11 materials-15-00970-f011:**
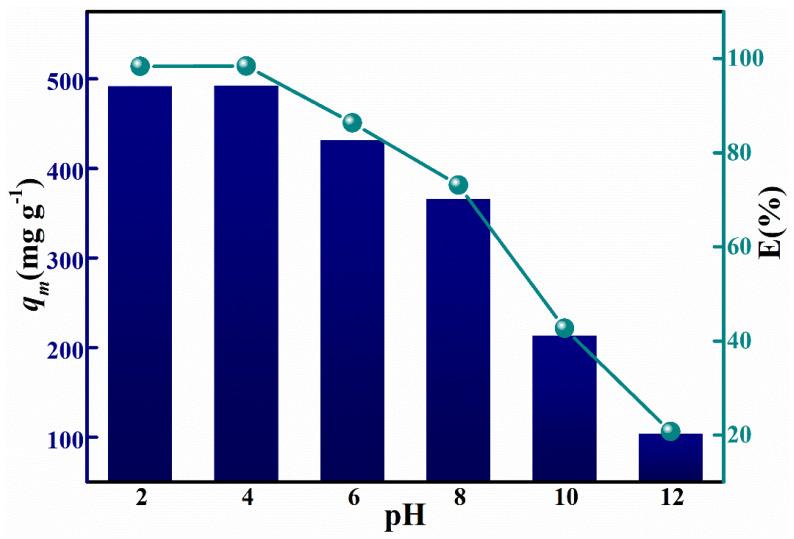
pH effect on the equilibrium adsorption capacity and removal efficiencies of CR in HPMA samples.

**Table 1 materials-15-00970-t001:** Approximate market price for HPA by product purity. Ranges are estimated from various sources.

HPA Purity (by Weight)	3N (99.9%)	4N (99.99%)	5N (99.999%)
**approx. market price range (US$/tonne**)	$5000–$15,000	$15,000–$30,000	$30,000–$50,000

**Table 2 materials-15-00970-t002:** Components of industrial aluminum hydroxide.

Component Content (*w*_B_/%)			
SiO_2_	Fe_2_O_3_	Na_2_O	Al(OH)_3_
1.25	1.16	0.58	97.01

**Table 3 materials-15-00970-t003:** Influence of initial NaAl(OH)_4_ solution concentration on impurity removal efficiency and alumina purity.

Concentration (g L^−1^)	Removal Efficiency (%)				Alumina Purity (%)
	Fe_2_O_3_	SiO_2_	Na_2_O	Total Removal Rate	
30	99.58	99.21	92.06	97.97	99.9393
60	99.44	99.57	96.21	98.86	99.9661
90	99.28	99.37	94.31	98.36	99.9509
120	99.22	99.33	93.45	98.15	99.9447

**Table 4 materials-15-00970-t004:** Influence of aging temperature on impurity removal efficiency and alumina purity.

Aging Temperature (°C)	Removal Efficiency (%)				Alumina Purity (%)
	Fe_2_O_3_	SiO_2_	Na_2_O	Total Removal Rate	
30	99.45	99.56	93.10	98.26	99.9481
60	99.44	99.57	96.21	98.86	99.9661
90	99.32	99.48	95.17	98.58	99.9576
120	99.30	99.52	94.83	98.53	99.9552

**Table 5 materials-15-00970-t005:** Effect of aging time on impurity removal efficiency and alumina purity.

Aging Time (h)	Removal Efficiency (%)				Alumina Purity (%)
	Fe_2_O_3_	SiO_2_	Na_2_O	Total Removal Rate	
1	99.42	99.55	93.45	98.31	99.9497
1.5	99.44	99.54	94.31	98.48	99.9547
2	99.44	99.57	96.21	98.86	99.9661
2.5	99.37	99.50	94.14	98.41	99.9526
3	99.32	99.42	92.41	98.03	99.9410
3.5	99.32	99.43	92.24	97.99	99.9401
4	99.32	99.44	91.72	97.89	99.9371

**Table 6 materials-15-00970-t006:** Effect of ammonium salt type on the impurity mass, alumina purity and porosity the product alumina.

Desodium Agent	Content (*w_B_*/%)				Specific Surface Area (m^2^ g^−1^)	Pore Volume (cm^3^ g^−^^1^)	Pore Size (nm)
	Fe_2_O_3_	SiO_2_	Na_2_O	Al_2_O_3_			
(NH_4_)_2_CO_3_	0.0065	0.0054	0.022	99.9661	312.43	0.43	3.80
CH_3_COONH_4_	0.0063	0.0057	0.054	99.9340	276.85	0.32	3.86
(NH_4_)_2_C_2_O_4_	0.0066	0.0054	0.050	99.9380	252.20	0.33	4.15

**Table 7 materials-15-00970-t007:** Effect of mass concentration of ammonium carbonate solution on impurity removal efficiency, specific surface area and alumina purity.

Concentration (g L^−1^)	Removal Efficiency (%)				Alumina Purity (%)	Specific Surface Are (m^2^ g^−1^)
	Fe_2_O_3_	SiO_2_	Na_2_O	Total Removal Rate		
25	99.44	99.54	93.28	98.28	99.9478	205.86
50	99.42	99.56	93.97	98.42	99.9528	276.54
75	99.44	99.57	96.21	98.87	99.9661	312.43
100	99.41	99.54	96.21	98.84	99.9653	311.59

**Table 8 materials-15-00970-t008:** Effect of roasting temperature on the specific surface area and pore volume.

Temperature	Specific Surface Area (m^2^ g^−1^)	Pore Volume (cm^3^ g^−1^)	Pore Diameter (nm)
500 °C	312.43	0.48	3.80
700 °C	199.35	0.36	5.36
900 °C	144.34	0.19	8.66
1100 °C	88.52	0.08	10.03

**Table 9 materials-15-00970-t009:** Impurity mass fraction in the product mesoporous alumina and its porosity.

Impurities	Fe_2_O_3_	SiO_2_	Na_2_O	Total Values	Specific Surface Area (m^2^ g^−1^)	Pore Volume (cm^3^ g^−1^)	Pore Size (nm)
Content (*w_B_*/%)	0.0065	0.0054	0.022	0.0339			
Remove rate (%)	99.44	99.57	96.21	98.87	312.43	0.48	3.80

**Table 10 materials-15-00970-t010:** Kinetic parameters for adsorption of CR on the HPMA powder.

*q_e-exp_* (mg g^−1^)	495		
**pseudo-first-oder model**	***q_e-cal_* (mg g^−1^)**	**K_1_ (min^−1^)**	** *R* ^2^ **
	51.36	0.03236	0.59214
**pseudo-second-oder model**	***q_e-cal_* (mg g^−1^)**	**K_2_ (min^−1^)**	** *R* ^2^ **
	495.05	0.00334459	0.99992

**K_1_** is the kinetic parameter fitted by the 1st-order model. **K_2_** is the kinetic parameter fitted by 2nd-order model.

**Table 11 materials-15-00970-t011:** Intra-particle diffusion model constants and correlation coefficients for adsorption of CR.

**K_d1_ (min^−1^)**	**C_1_**	** *R* _1_ ^2^ **
247.343	−1.82167	1
**K_d2_ (min^−1^)**	**C_2_**	** *R* _2_ ^2^ **
73.47937	254.92449	0.95626
**K_d3_ (min^−1^)**	**C_3_**	** *R* _3_ ^2^ **
2.11518	470.82288	0.82684

**K_di_** is the value of slope fitted by I-D model; **C_i_** is the corresponding intercept value.

**Table 12 materials-15-00970-t012:** Adsorption isotherms parameters for adsorption of CR on the HPMA powder.

**Langmuir model**	***q_m_* (mg g^−1^)**	**K_L_**	** *R* ^2^ **
	2053.818	0.0476	0.99972
**Freundlich model**	**K_F_ (mg^1−(1/n)^ L^1/n^g^−1^)**	**n**	** *R* ^2^ **
	259.755	2.882	0.86148
**Temkin model**	**K_T_ (L mg^−1^)**	**b**	** *R* ^2^ **
	1.021	7.1853	0.97039

**Table 13 materials-15-00970-t013:** Comparison of the maximum monolayer adsorption (*q_m_*) of different adsorbents for the removal of CR.

Adsorbents	*q_m_*(mg g^−1^)	References
	CR	
**HPMA nanopowder**	1984.64	This study
**a-Fe_2_O_3_ nanoparticles and nanowhiskers**	254.00	[68]
**HAM@ γ-AlOOH/Fe(OH)_3_**	252.53	[69]
**Fe_3_O_4_@P4-VP nanospheres**	151.50	[70]
**spindle-like γ-Al_2_O_3_**	176.70	[71]
**α-FeOOH nanorods**	104.2	[72]
**nanorod-like mesoporous γ-Al_2_O_3_**	83.80	[73]
**magnetic core-manganes oxide shell**	43.00	[74]
**Fe_2_O_3_@mSiO_2_**	72.22	[65]

## Data Availability

The data used to support the findings of this study are available from the corresponding author upon request.

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
