# Peer review of "Synthesis and Characterization of High-Purity Mesoporous Alumina with Excellent Adsorption Capacity for Congo Red"

_materials, 2022, doi:10.3390/ma15030970_

Round 1
Reviewer 1 Report
Authors prepared high purity mesoporous alumina, optimizing the synthesis route. The obtained materials have been analyzed by electron microscopy, nitrogen sorption and XRD, and finally tested for adsorption of Congo Red organic pollutant. The kinetics of sorption has been evaluated by three commonly used models, two of which did not fit the data at all, and the third one, the pseudo second order kinetics exhibited good fitting. It should be noticed however, that the apparent agreement does not show the true details of the sorption process, because it is shadowed by the multiplication of the (1/qt) term by the time t, which is in the same time the independent variable of the x axes. This has been discussed in recent literature.
The data plotted in figure 10a indicate that the saturation is reached within 20 minutes. Therefore, only the first 20 minutes are relevant for testing the various kinetic models. Porperly chosen data sampling intervals, more frequent in the first 20 min, and replica experiments should be performed to obtain accurate data and reveal the possible sorption kinetics.
The second weakness of the work is, that no sorption isotherm data have been taken for Congo Red, and no thermodynamic analysis of the dye sorption mechanism. Such data and analysis would be highly appreciated.
These items can be revised in the revised version of the paper.
Author Response
Thank you for your comments on our manuscript, and the related revision has been made in the manuscipt, the details are shown in the attached document.

Reviewer 2 Report
This manuscript contains a new material on synthesis and characterization of high purity mesoporous alumina, but not so much on absorbtion activities on Congo red. It is nessary to add data on comparision with anderes adsorbents in this field.
- To my mind the main subject of this research is to develop new adsorbent with adsorbtion properties for removal Congo red from water solutions.
- Material is new and unteresting and relevant for solving some problem of obtaining pure technical waters.
- Topic of this manuscript is original in direct ageing and ammonium salt substitution methods.
- Paper is written well, but conclusions are not clear. It is nessasary to write new conclusions without same data on experiments, only results. Text is clear and easy to read.
- It is new applications of mesoporous alumina as adsorbents.
Author Response

(The authors gave the same response as above.)

Reviewer 3 Report
Journal: Materials
Title: Synthesis and characterization of high purity mesoporous alumina with excellent absorption property of Congo red“ – absorption or adsorption
The authors present the synthesis of HMPA and its application for the removal of Congo Red dye. In my opinion, the paper is extremely unsystematically written in all its parts. It takes a really major major revision to be considered for possible publication at all. I will suggest rejecting in this form and re-submitting or putting in a lot of effort, literally re-writing the paper. I will explain my opinion in detail. First, it appears that the authors did not read at all or in detail the latest version of their paper before submission. Chapter 3.2 is missing; Figure 9 is missing. In fact, it is not missing but skipped. Finally, they mention the "adsorption process of Congo red on synthetic magnesium oxide" which is unacceptable because there is no mention of MgO in the paper. It's like copying from a previous manuscript. Chapter 3 and all subchapters begin with a Figure and Table, without previous introductory text and announcements of Figures and Tables. In Chapter 3.4. model equations appear after figures and tables. All of these are technical problems that create confusion. See other technical issues in specific comments.
The introductory part is unconnected. First, different methods of HPA synthesis are mentioned, then Congo red, and then the authors return, but now to HPMA synthesis.
Why is it important to explain different HPA syntheses? For the synthesis of HPMA, you used industrial Al (OH) 3 as written in the experimental part. Is it a purchased chemical or really from some industry? I am interested in the importance of methods of obtaining HPA?
Then you mention the Congo red, its use and toxic properties. That is OK. It should refer to the real problem related to CR in your country. Why research something, I guess to possibly apply it. How CR is being removed in the industry's wastewater so far. What is the concentration legislation for CR in wastewater or COD value?
Why did you choose the concentration range for CR listed in the experimental section?
Chapter 2, the experimental part needs to be completely revised, written and explained from the beginning, especially 2.2. and 2.4. For example, from subsection 2.2. I realized nothing, and later in Chapter 3 I realized that you were examining different experimental conditions for HPMA synthesis. Please explain in detail the implementation of the experiment in Chapter 2. Also in Chapter 3, please do not describe the experimental conditions and the implementation of the experiment in the text or Figure captions! In subsection 2.4. you mention the concentration range for CR 50-250mg / L, and the mass of the adsorbent of 100 mg. Further in the results, you use a concentration of 250 mg / L for CR, kinetic tests and an adsorbent mass of 500 mg, in some parts of the text, respectively 100 mg. Confusing!
Related to Chapter 3, more precisely all the subsections of Chapter 3.1: - This is the worst part of the paper. All the time, the removal of impurities from the initial sample Al (OH)3, which has the purity of the main component 97.01 with minor, Na, Si and Fe (results from Table 1), is emphasized. In my opinion, this is unacceptable, I would just present the results in the opposite way, emphasizing how much I got the purity of Al (OH) 3 or synthesized HPMA during different synthesis conditions, aging, temperature, etc.
Table 2:
the effect of concentration is negligible on the removal of Fe and Si, relative to possibly Na. Removal rate what does it apply to? Show purity, Al (OH) 3 content and percentage, it is important - how pure it is. To discuss the removal of Si and Fe in the order of 0.00x is insignificant. Also in the whole chapter, the Tables and the Figures that show the same, the Tables are more important. By the way, here I realized that you performed the impact of different concentrations.
Table 3:
Same as before - product purity on Al in HPMA.
Column 1 shows the different temperatures. The concentration of NaAl (OH) 4 is not observed.
Table 4:
Same as before - product purity on Al in HPMA.
Column 1 shows the different aging time. The concentration of NaAl (OH) 4, as well as the temperature at which it is carried out are not observed.
Table 5:
Same as before - product purity on Al in HPMA.
Column 1 shows the different NH4-salts. The concentration of NaAl(OH)4, temperature as well as aging time were not observed. In Tables 2-4, which NH4 salts were used?
Table 6, 7 and 8, emphasize other experimental conditions as well as purity on Al in HPMA.
Chapter 3.3. does not have an appropriate name, this is about highlighting the most optimal procedure. There is also a lot of repetition in the descriptions. Be more concise.
Chapter 3.4. - The intraparticle diffusion model is actually a Weber Morris model that is linearized into two parts, consulting the literature. Since it is a porous material, it is certainly a combined mechanism of mass transfer.
Explaining the model is trivial. It should be used to define a possible mechanism of CR adsorption on HPMA. I am interested in the pH of real wastewater containing CR, and why the authors chose pH = 4. Also define the reaction mechanism, write, explain by chemical reaction. The results of the zeta potential of the prepared material would also serve this purpose.
Conclusion - very weak
SPECIFIC COMMENTS:
TITLE: „Synthesis and characterization of high purity mesoporous alumina with excellent absorption property of Congo red“ – absorption or adsorption??? I think adsorption.
„# Zhonglin Li and Ding Wang contributed equally to this work“ - The contribution of each author is added according to the template at the end of this paper. Look at the template and compile the next version of the paper according to the template. According to what you have written, it seems that only the two authors mentioned participated in the preparation of the paper, and the others ....
ABSTRACT:
„In order to solve the problem that some conventional micro and nano adsorbents have difficulty in showing the adsorption sites of adsorbed pollutants in solution due to heavy aggregation of the adsorbent while exploring a more concise process for the preparation of high purity alumina.“ - The meaning of the sentence!
Key words: „ CR adsorption“ -Write CR with full name.
INTRODUCTION
„High purity mesoporous alumina (HPMA) with its large specific surface, suitable pore structure, excellent physical properties, chemical stability and activity has been extensively employed in the field of Congo red adsorption.“
- EXPERIMENTAL PROCEDURES
2.2 Preparation of High-purity Mesoporous γ-Al 2 O 3
NaAl(OH)4 solution – 4 in subscript
„then aged at appropriate temperature for a period of time.“ - Start a sentence with a capital letter. How much time?
„in a blast drying oven for more than 2 h, and then roasted (rising rate of 10 ºC/min)“ -Dried at temperature? Roasted at temperature?
2.3 Characterization of Alumina Materials
„BET and BJH methods respectively“ - Although these methods are well known to researchers involved in material characterization, I would still use the full name, then the abbreviation if necessary.
„The concentration of CR in the solution was determined using a Shimadzu UV-9000S spectrophotometer.“ - This sentence does not belong to this chapter, characterization of the material.
2.4 NPMA materials Degradation of CR - Do not use abbreviations in the title or text if they are mentioned for the first time and are not a previous explain with the full name.
„The absorption“ or adsorption
„have been carried by degradation of CR.“ -It is not clear to me "degradation". Is CR degradation occurring or is it sorption on the material?
„adjusted to the desired pH“ – which pH?
- Results and Discussion
3.1. Effects of various factors on the MA product – again abbreviation in title
3.1.1 Initial concentration of NaAl(OH) 4 solution -A chapter cannot start with a Table without the previous introductory text to announce the Table. Same regarding Figure 1.
3.1.6 Roasting temperature
„the product phase is γ-Al 2 O 3 phase transformation when roasted at 900 °C,…“ – or from 500-900°C is γ-Al2O3
Chapter 3.2 is missing.
3.3 Preparation of high purity mesoporous alumina
„NH 4+ can be absorbed by γ-AlO(OH), “ – or adsorbed?
„The impurity mass ratios, excellent removal rates and surface physical properties of“ - removal rates, what, what does it refer to ??
„which can be widely used in the preparation of catalysts“ – catalysts??
Figures 8a and e are the repeated curves of Figure 6 at 500 ° C? I would say they are repeated.
Figure 8d shows a similar detail shown as in Figure 6a, only the second detail. Unnecessary repetition of the same.
Figure 9 is missing!
„was fixed at 250 mg L -1 and the amount of adsorbent was 100 mg“ - The explanation of the title of Figure 10 states that the adsorbent concentration is 500 mg / L ??????
„which can be widely used for the degradation of CR in“ – again degradation
„adsorption process of Congo red on synthetic magnesium oxide is very consistent“ - magnesium oxide, This is completely frivolous.
Author Response

(The authors gave the same response as above.)

Round 2
Reviewer 1 Report
Authors performed a substantial revision of the paper, however it is very difficult to read it because of the tracked changes. In a new version, it would be much better to indicate the changed parts by color highlighting, and omit the deleted parts of the original manuscript.
1. Many problems of the use of pseudo-second order kinetics are shown in this paper:
https://www.sciencedirect.com/science/article/pii/S266679082030032X
The many data points that are measured after the saturation, do not contribute to the information content of the data, and often mask the otherwise poor fit in the short time part of the data.
In the revised version, authors presented repeated kinetic measurements, with much more closely spaced points in the beginning. They apparently fall on a straight line, supporting the suggested second order kinetics. However, the same data plotted in the original coordinates, (figure R2 a) show a marked sharp kink at around 10 minutes. Could the authors please explain this phenomenon?
2. Authors performed isotherm measurements and concluded about the Langmuir type of dependence. However, the used concentration range and sorbent concentration actually prevented them to achieve isotherms with clear sign of saturation of q_eq in the function of C_eq . (Figure 10.a.) Note,that in this figure the maximum C_eq was only 20 mg/L, which is much below the initial maximum concentration of CR.
Proper combination of the adsorbent amount, and the sorbate concentrations should be used to achieve isotherm plots which show saturation, see for example:
https://doi.org/10.1016/j.apsusc.2013.11.108
https://doi.org/10.1016/j.chemosphere.2020.127737
for Congo Red.
The presented data (Figure 10) therefore do not allow accurate determination of the adsorption capacity, this is probably the reason of the unusually high values obtained. In addition, the 2 gram CR on 1 gram Al support cannot be imagined to form a monolayer, as hypothesized by the Langmuir model.
Revisit of this part, and further adsorption measurements are necessary to support the Langmuir mechanism and accurately determine the adsorption capacity. Please use non-linear fits for the various isotherms, and plot them in the original coordinates, which would allow an easy visual verification of the correctness of the chosen isotherm model.
3. The table presenting comparison with other sorbents for CR may be completed by popular magnetic silica sorbents, such as https://doi.org/10.1016/j.chemosphere.2020.127737
Author Response
Thank you for your review of our paper and for your criticisms and related corrections, which has been revised and explained in the attatched file.

Reviewer 3 Report
The manuscript has been corrected, but not enough. Some important questions remained open. I would like to read corrected manuscript once again in detail .
I ask the authors to clearly mark all corrections in the text (other letter color) as well as in the answer.
PREVIOUS COMMENT
Chapter 3.2 is missing.
NEW COMMENT
Chapter 3.2 is missing. Carefully check the number of tables and chapter numbering!
Line 616: 3.3. change to 3.4
PREVIOUS COMMENT
Then you mention the Congo red, its use and toxic properties. That is OK. It should refer to the real problem related to CR in your country. Why research something, I guess to possibly apply it. How CR is being removed in the industry's wastewater so far. What is the concentration legislation for CR in wastewater or COD value?
Why did you choose the concentration range for CR listed in the experimental section?
NEW COMMENT
What is the concentration legislation for CR in wastewater or COD value?
Why did you choose the concentration range for CR listed in the experimental section?
I did not get answers to these two questions. Include the answer in the manuscript.
PREVIOUS COMMENT
Also in Chapter 3, please do not describe the experimental conditions and the implementation of the experiment in the text or Figure captions!
NEW COMMENT
Not corrected. Since the authors through chapter 3.1. wrote in detail the experimental conditions, it may be good to stay in the discussion. The only question is whether the performance of the experiment is well enough explained.
Line 186: „To study the adsorption thermodynamics,….“ – thermodynamics??????
Line 196: „m: Mass of NPMA materials (g)“ – NPMA?????
I think that the answer given by the authors to me regarding the cost of HPA depending on the purity (3N, 4N ..) is extremely important. Therefore, I think it is important for the authors to devise a way to incorporate the answer into the manuscript in at least one sentence. Why, because with this answer, the discussion makes sense, and without it it does not, since I referred in the review to this very issue, the importance of product purity, and on your part the importance of removing impurities.
Lines 594-602: Describe the determination of zeta potential in Chapter 2.2. At what pH is the zeta potential determined?
PREVIOUS COMMENT
Chapter 3.3. does not have an appropriate name, this is about highlighting the most optimal procedure. There is also a lot of repetition in the descriptions. Be more concise.
ANSWER: It has been revised, thanks a lot.
NEW COMMENT
I didn't notice any corrections, just adding zeta potential results.
PREVIOUS COMMENT
Chapter 3.4. - The intraparticle diffusion model is actually a Weber Morris model that is linearized into two parts, consulting the literature. Since it is a porous material, it is certainly a combined mechanism of mass transfer.
NEW COMMENT
I disagree with the answer. I clearly notice two linear parts. See literature.
PREVIOUS COMMENT
Explaining the model is trivial. It should be used to define a possible mechanism of CR adsorption on HPMA. I am interested in the pH of real wastewater containing CR, and why the authors chose pH=4. Also define the reaction mechanism, write, explain by chemical reaction. The results of the zeta potential of the prepared material would also serve this purpose.
NEW COMMENT
The authors did not answer any questions in the answer letter! They also did not make any changes to the manuscript required from the above questions! Zeta potential results were added, which did not serve in defining the mechanism.
Line 697: „thermodynamic data.“ – thermodynamic??????
Lines 702-703: „sorption process is more in line with the Langmuir isotherm model,“
And
Lines 705-706: „And the maximum adsorption capacity calculated with was calculated Langmuir isotherm model is 1984.560 mg g -1 .“
According to Figure 9a, the experimentally obtained capacity is 492.12 mg/g, and calculated according to the Langmuir isotherm is 1984.560 mg/g. I wonder how then the Langmuir isotherm fits the experimental data well?
Lines 716-717: „and its theoretical adsorption maxima for Congo Red calculated with Langmuir Model were compared with other as-reported metal hydroxides.“ – your experimentally obtained capacity is 492.12 mg/g, not 1984.560 mg/g as stated in Table 2 (or in line 718 Table R2, or now Table 11)
Lines 715-723: These statements are debatable.
Correct the conclusions in accordance with the new corrections in the manuscript.
PREVIOUS COMMENT
Zhonglin Li and Ding Wang contributed equally to this work”-The contribution of each author is added according to the template at the end of this paper. Look at the template and compile the next version of the paper according to the template. According to what you have written, it seems that only the two authors mentioned participated in the preparation of the paper, and the others ....
As per your request, the full text has been formatted according to the template and specific work done by all authors for this study has been added, thanks for your tips.
NEW COMMENT
This statement should be at the end of the paper. See instructions for authors.
PREVIOUS COMMENT
Introduction: “High purity mesoporous alumina (HPMA) with its large specific surface, suitable pore structure, excellent physical properties, chemical stability and activity has been extensively employed in the field of Congo red adsorption.”
It only mentions the content of the sentence, but what exactly is wrong with it is not described, so I am so sorry that we have not revised the sentence.
NEW COMMENT
I apologize for not commenting. Please cite relevant literature supporting CR adsorption studies on HPMA.
Author Response

(The authors gave the same response as above.)
